# Correlates of SARS-CoV-2 Breakthrough Infections in Kidney Transplant Recipients Following a Third SARS-CoV-2 mRNA Vaccine Dose

**DOI:** 10.3390/vaccines13080777

**Published:** 2025-07-22

**Authors:** Miriam Viktov Thygesen, Charlotte Strandhave, Jeanette Mølgaard Kiib, Randi Berg, Malene Söth Andersen, Emma Berggren Dall, Bodil Gade Hornstrup, Hans Christian Østergaard, Frank Holden Mose, Jon Waarst Gregersen, Søren Jensen-Fangel, Jesper Nørgaard Bech, Henrik Birn, Marianne Kragh Thomsen, Rasmus Offersen

**Affiliations:** 1Department of Clinical Microbiology, Aarhus University Hospital, 8200 Aarhus, Denmark; msan@clin.au.dk (M.S.A.); marthoms@rm.dk (M.K.T.); 2Department of Medicine, Gødstrup Hospital, 7400 Herning, Denmark; jeanettekiib@gmail.com (J.M.K.); emjoer@rm.dk (E.B.D.); bodil.hornstrup@rm.dk (B.G.H.); hansos@rm.dk (H.C.Ø.); rasmus.offersen@gmail.com (R.O.); 3Department of Renal Medicine, Aarhus University Hospital, 8200 Aarhus, Denmark; henrbirn@rm.dk; 4Department of Nephrology, Aalborg University Hospital, 9000 Aalborg, Denmark; charlotte.strandhave@rn.dk (C.S.); jon.gregersen@rn.dk (J.W.G.); 5Department of Clinical Medicine, Aalborg University, 9000 Aalborg, Denmark; 6Department of Clinical Immunology, Aarhus University Hospital, 8200 Aarhus, Denmark; randi.berg@rm.dk; 7Department of Clinical Medicine, Aarhus University, 8000 Aarhus, Denmark; frank.holden.christensen@goedstrup.rm.dk (F.H.M.); jesper.noergaard.bech@goedstrup.rm.dk (J.N.B.); 8Department of Clinical Epidemiology, Aarhus University, 8200 Aarhus, Denmark; 9University Clinic in Nephrology and Hypertension, Department of Medicine, Gødstrup Hospital, 7400 Herning, Denmark; 10Department of Infectious Diseases, Aarhus University Hospital, 8200 Aarhus, Denmark; soejense@rm.dk

**Keywords:** COVID-19, machine learning predictors, transplant immunology, immunosuppression

## Abstract

Background: Kidney transplant recipients (KTRs) exhibit a significantly diminished immune response to Severe Acute Respiratory Syndrome Coronavirus-2 (SARS-CoV-2) vaccines compared with the general population, primarily due to ongoing immunosuppressive therapy. This study evaluated the immunogenicity of a third SARS-CoV-2 mRNA vaccine dose in KTRs and assessed the association between antibody response and protection against SARS-CoV-2 breakthrough infection. Additionally, the clinical and immunological correlates of post-vaccination SARS-CoV-2 infection were examined. Methods: A prospective cohort of 135 KTRs received a third vaccine dose approximately six months following the second dose. Plasma samples were collected at baseline (pre-vaccination), six months after the second dose, and six weeks following the third dose. Humoral responses were assessed using SARS-CoV-2-specific Immunoglobulin G (IgG) titers and virus neutralization assays against wild-type (WT) and viral strains, including multiple Omicron sub-lineages. Results: After the third vaccine dose, 74% of the KTRs had detectable SARS-CoV-2-specific IgG antibodies, compared with 48% following the second dose. The mean IgG titers increased approximately ten-fold post-booster. Despite this increase, neutralizing activity against the Omicron variants remained significantly lower than that against the WT strain. KTRs who subsequently experienced a SARS-CoV-2 breakthrough infection demonstrated reduced neutralizing antibody activity across all variants tested. Additionally, individuals receiving triple immunosuppressive therapy had a significantly higher risk of SARS-CoV-2 breakthrough infection compared with those on dual or monotherapy. A multivariate machine learning analysis identified age and neutralizing activity against WT, Delta, and Omicron BA.2 as the most robust correlates of SARS-CoV-2 breakthrough infection. Conclusions: A third SARS-CoV-2 mRNA vaccine dose significantly improves SARS-CoV-2-specific IgG levels in KTRs; however, the neutralizing response against Omicron variants remains suboptimal. Diminished neutralizing capacity and intensified immunosuppression are key determinants of SARS-CoV-2 breakthrough infection in this immunocompromised population.

## 1. Introduction

The immune response to Severe Acute Respiratory Syndrome Coronavirus-2 (SARS-CoV-2) messenger ribonucleic acid (mRNA) vaccines is impaired in solid organ transplant recipients (SOTRs) compared with that in the general population [1,2]. Immunocompromised patients, including kidney transplant recipients (KTRs), are considered at higher risk of severe COVID-19 than the general population, with increased mortality [3,4]. A third SARS-CoV-2 mRNA vaccine dose may provide crucial additional protection in this vulnerable population.

The development of SARS-CoV-2 vaccines represents a significant breakthrough in combating SARS-CoV-2 transmission. However, constant, rapid viral mutations and the emergence of multiple SARS-CoV-2 variants pose ongoing challenges [5]. Furthermore, fully vaccinated KTRs exhibit low rates of SARS-CoV-2 anti-spike Immunoglobulin G (IgG) seroconversion as well as lower IgG titers and neutralizing abilities [2,6]. Immunosuppressive regimens, as well as patient-related factors, including age and comorbidities, have been identified as potential independent risk factors for a diminished antibody response in SOTRs [7].

A previous study investigating the predictive factors for the immune response to a third SARS-CoV-2 mRNA vaccine dose based on acquired neutralizing capacity of the receptor-binding domain (RBD) IgG showed that the responders had a longer time since transplantation and were less frequently exposed to mycophenolate mofetil and tacrolimus. Also, responders with pre-existing non-neutralizing anti-RBD IgG or a positive Interferon Gamma Release Assay (IGRA) test before the third vaccine dose showed a better response after the third vaccine dose [8].

While antibody levels often correlate with neutralizing activity, their association with SARS-CoV-2 breakthrough infections in SOTRs is not well studied. This is particularly relevant since antibodies exhibit varying neutralizing activity depending on the viral variant. A large multicenter study found a dose-dependent association between SARS-CoV-2 anti-spike IgG levels after a third vaccine dose and the risk of SARS-CoV-2 breakthrough infection with the Delta variant but not with Omicron [9].

A deeper understanding of the predictive factors influencing the immune response to SARS-CoV-2 vaccination, as well as the correlates of protection against SARS-CoV-2 breakthrough infection, is crucial for preventing disease transmission and developing next-generation vaccines for immunocompromised patients.

This prospective, observational study examined the immune response to a third SARS-CoV-2 mRNA vaccine dose in KTRs and assessed antibodies as correlates of protection along with other potential correlates of SARS-CoV-2 breakthrough infection.

## 2. Materials and Methods

### 2.1. Participants

In this prospective, observational, multicenter cohort study, we examined 135 kidney recipients for SARS-CoV-2-specific IgG and neutralizing antibody activity. Adult KTRs (>18 years of age) were identified from three Danish centers: Department of Medicine, Gødstrup Hospital; Department of Renal Medicine, Aarhus University Hospital; and Department of Nephrology, Aalborg University Hospital from 15 January 2021 to 21 February 2022. They were recruited either by electronic letter, as a part of routine outpatient visits, or on attendance to receive the first vaccine dose. The participants were enrolled before they received a BNT162b2 (Pfizer Inc., New York, NY, USA and BioNTech^®^, Mainz, Germany) or mRNA-1273 (Moderna, Cambridge, MA, USA) vaccine as part of a routine national immunization program independent of this study. Vaccinations with doses 1–3 were carried out from January 2021 to October 2021. In brief, the first two vaccine doses were administered with a median interval of 22 (range 20–44) days. Participants received their third vaccine dose 183 (median; range 133–207) days following the second dose. Plasma samples were obtained from August 2021 to January 2022, at baseline (pre-vaccination), 6 months after the second dose, and 6 weeks following the third dose. A timeline of the vaccine dose administration and the sampling is shown in Appendix A. The patient data was recorded from the patient record files and included demographics, time since transplantation, immunosuppressive regimens, and SARS-CoV-2 polymerase chain reaction (PCR) results.

SARS-CoV-2 breakthrough infection was defined as a positive SARS-CoV-2 PCR analysis after the third vaccine dose with follow-up until 21 February 2022. Specific SARS-CoV-2 variants were not registered systematically, because the KTRs were tested at different locations. However, the plasma sampling before and after the third vaccine dose occurred when the Delta variant was dominating in Denmark, which was also just before a major surge in Omicron-driven COVID-19 cases in Denmark, peaking in February/March 2022. We therefore assume that the SARS-CoV-2 breakthrough infections were predominantly Omicron variants. See Methods 2.6 for more details.

To avoid bias from hybrid or passive immunity, patients who had been infected with SARS-CoV-2 and/or were receiving SARS-CoV-2-specific monoclonal antibodies before the last plasma samples were obtained were excluded from the analysis. We also excluded KTRs who were lost to follow-up including death or discontinued immunosuppression therapy due to nephrectomy. All the KTRs included followed the vaccination schedule and had plasma samples collected as described. A flow chart of the inclusion and exclusion of participants is shown in Figure 1.

### 2.2. SARS-CoV-2 Antibody Measurement

For antibody quantification, EDTA tubes were centrifuged, and the plasma was subsequently stored at −80 °C. Anti-spike SARS-CoV-2 IgG was detected using the LIAISON^®^ SARS-CoV-2 TrimericS IgG assay (Diasorin SA, Antony, France) on a Liaison XL fully automated chemiluminescence analyzer (Diasorin SA, Antony, France) [10]. The spike protein is a trimeric complex of glycoprotein consisting of three protomers, each with two subunits (S1 and S2). In the S1 subunit, there is a receptor-binding domain (RBD) and an N-terminal domain (NTD), which are the most immunogenic regions. The test detects IgG antibodies against the trimeric complex, including the RBD and NTD sites from the three S1 subunits. Quantitative results are expressed as binding antibody units (BAU/mL). A definitive positive threshold was defined as ≥33.8 BAU/mL.

The assay’s performance characteristics, as reported by Bonelli et al. [10], include a clinical sensitivity of 98.7% (≥15 days after a positive PCR result) and a specificity of 99.5% (95% confidence interval (CI) 99.0–99.7%). The intra-assay coefficient of variation (CV) is 1.6–5.1%, and the inter-assay CV is up to 6% according to the kit insert. Calibrator concentrations, expressed in BAU/mL, have been standardized by the manufacturer against the First WHO International Standard for anti-SARS-CoV-2 immunoglobulin (NIBSC code: 20/136). All the samples were analyzed by experienced laboratory technicians in accordance with the manufacturer’s protocol, including the daily testing of the LIAISON^®^ SARS-CoV-2 TrimericS IgG internal control samples. The laboratory analysis was blinded to clinical outcomes.

### 2.3. Antibody Neutralization

Antibody neutralizing activity against the SARS-CoV-2 variants was analyzed using an angiotensin-converting enzyme 2 (ACE2) pseudo-neutralization assay as described earlier [11]. Briefly, an MSD^®^ immunoassay (V-PLEX SARS-CoV-2 Panel (ACE-2) Kit; cat. number K15586U) was used to measure the ability of plasma samples to inhibit (ACE2) binding to different variants of SARS-CoV-2 spike including wild-type SARS-CoV-2 spike (WT), B.1.1.7/Alpha, AY.4/Delta, B.1.351/Beta, BA.1/Omicron, BA.1+R346K/Omicron, BA.1+L425R/Omicron, BA.2/Omicron, and BA.3/Omicron. The 96-well MSD^®^ plates were blocked with MSD^®^ Blocker for 30 min. The plates were then washed in MSD^®^ washing buffer, and 25 μL of each plasma sample was added to the plates. The plates were incubated for 1 h, and recombinant human ACE2-SULFO-TAG™ was added to the wells. After an additional 1 h, the plates were washed, and MSD GOLD™ Read Buffer B was added. The plates were read immediately using a MESO^®^ QuickPlex SQ 120MM reader (Danaher Corporation, Washington, DC, USA). The neutralizing activity was determined by measuring the presence of antibodies able to block the binding of ACE2 to SARS-CoV-2 spike proteins from WT spike, B.1.1.7/Alpha, B.1.617.2/Delta, B.1.351/Beta, P.1/Gamma, and Omicron BA.1 and reported as the percentage of ACE2 inhibition in comparison to samples with no inhibitory antibodies added on the same plate. The laboratory analysis was blinded to clinical outcomes.

### 2.4. Statistical Analysis

Paired comparisons of the median antibody levels before and after the third SARS-CoV-2 mRNA vaccine dose were performed using a Mann–Whitney test. The antibody-mediated neutralization against different variants of SARS-CoV-2 was compared with that against the WT strain using a Kruskal–Wallis test and Dunn’s multiple comparison test post hoc. Antibody-mediated neutralization was compared between patients with and without breakthrough infection using the Mann–Whitney test for each variant of SARS-CoV-2. A threshold of *p* < 0.05 was used to define a statistically significant result. Fisher’s Exact Test was used to assess the association between the immunosuppressive regimens (triple vs. non-triple therapies) and the SARS-CoV-2 breakthrough infections. The analyses and the graphing were carried out using GraphPad Prism 10.3.1 and JMP Pro v. 16 and RStudio Version 2024.12.0+467.

### 2.5. Machine Learning Models

Correlates of the SARS-CoV-2 breakthrough infection were identified using a two-step machine learning model combining elastic net regression followed by Bootstrap Forest [12,13] using 5-fold cross-validation in each model.

Data Processing: The raw data were analyzed using JMPPro 16 (SAS Institute). Features with >20% missingness were excluded; the remaining missing values were imputed with k-nearest neighbors (k = 5).

Feature Selection: Elastic net, which applies a dual penalty on both the absolute values of coefficients and the sum of squared coefficients, enabling the algorithm to shrink the coefficients toward zero, was first used to prioritize important features. We loaded 16 features based on antibody levels, neutralizing activity for each variant, immunosuppressant regimens, time since transplantation, and age.

On the training set (n = 107), we fit an elastic net model (α = 0.5) via 5-fold cross-validation to identify predictive variables. The penalty λ minimizing the mean cross-validation error retained 5 nonzero terms for subsequent modeling in the Bootstrap Forest model. Early stopping was used.

Normalization and Scaling: All the continuous predictors were standardized; the categorical predictors were dummy-coded. Standardization ensured comparability both under the elastic net penalty and for the random-forest split criteria.

The model selected 5 features, and for impartiality testing, these were next applied to an independent Bootstrap Forest model. Bootstrap Forest is an ensemble learning method that constructs multiple decision trees by randomly sampling the dataset with replacement. Each tree is trained on a different subset of the data, and the final predictors are determined by aggregating predictions from all the individual trees. The predictors identified in the training set are then tested on a subset of data, different from the training set, using a 5-fold cross-validation. Finally, the model is applied on a test dataset not used in the model. The area under the curve is reported for both the cross-validated training set and the test set.

### 2.6. National COVID-19 Infection Rates

To better understand and illustrate the findings of the study in the context of the evolving COVID-19 pandemic on a national level, we collected information about COVID-19 infection rates and SARS-CoV-2 variants of concern (VOC) in the general population in Denmark during the period from January 2021 to November 2022. The data was obtained on 15 June 2024 from the publicly available webpage of the Danish national health institute, ‘Statens Serum Institute’, and from interactive dashboards showing the total number of COVID-19 infections in Denmark over time [14] and proportions of SARS-CoV-2 VOC dominating in Denmark [15]. We modified the figures with permission by combining the two graphs from the interactive dashboards with our study timeline illustrating vaccine dose administrations and data sampling (Appendix A).

## 3. Results

### 3.1. Clinical and Demographic Characteristics

Of the 135 KTRs included, 131 received the BNT162b2 and 4 the mRNA-1273 vaccine. The baseline characteristics are shown in Table 1.

The steroid-free dual-therapy regimens predominantly include the combinations of calcineurin inhibitor (CNI) + antimetabolite or sirolimus + antimetabolite. The antimetabolite-free dual-therapy regimens predominantly include CNI + prednisolone or sirolimus + prednisolone. The CNI-free regimens predominantly include sirolimus + antimetabolites or sirolimus + prednisolone. A Venn diagram showing all the combinations can be found in Appendix A.

### 3.2. Vaccine-Induced SARS-CoV-2 Antibody Response

Forty-eight percent of the KTRs were seropositive after the second vaccine dose when compared with baseline. After receiving the third vaccine dose, the rate of seropositive KTRs increased to 74% (Figure 2A).

Not only did the seropositive rate increase, but the levels of SARS-CoV-2-specific IgG also increased approximately ten-fold from 137 BAU/mL to 1396 BAU/mL among vaccine responders (*p* < 0.0001; Mann–Whitney test; Figure 2B). Thus, the KTRs had significantly higher SARS-CoV-2 antibody levels after the third vaccine dose.

### 3.3. Neutralizing Antibodies to Different SARS-CoV-2 Variants and Associations with SARS-CoV-2 Breakthrough Infection

Compared with the WT strain, the neutralizing activity of patient antibodies was significantly lower against all the Omicron variants examined (*p* < 0.0001; Kruskal–Wallis followed by Dunn’s) but not against any of the other variants: Alpha (B.1.1.7), Beta (B.1.351), or Delta (AY.4). This indicates severely diminished protection against Omicron variants (Figure 3).

Of the 135 KTRs, 25 had a SARS-CoV-2 breakthrough infection after the third vaccine dose. KTRs with SARS-CoV-2 breakthrough infection had significantly lower neutralizing activity against the WT strain (*p* = 0.005), as well as against the variants Alpha (*p* = 0.008), Beta (*p* = 0.002), Delta (*p* = 0.002), Omicron BA.1+L425R (*p* = 0.032), and Omicron BA.2 (*p* = 0.041) (Mann–Whitney test) (Figure 4). Thus, the antibody-mediated neutralizing activity was inversely associated with the SARS-CoV-2 breakthrough infections.

### 3.4. Immunosuppressive Regimen and Risk of SARS-CoV-2 Breakthrough Infection

KTRs on triple therapy (CNI + antimetabolite + steroid) had a significantly higher risk of a SARS-CoV-2 breakthrough infection when compared with those not on triple therapy (OR 2.63 [1.04–6.64]) (Fisher’s Exact Test). Indeed, SARS-CoV-2 IgG levels were lower in this group compared with those in KTRs not on triple therapy, with mean IgG levels of 225 BAU/mL and 1290 BAU/mL, respectively. No significant association with the risk of SARS-CoV-2 breakthrough infection was found among the other immunosuppressive regimens examined (Table 2).

The relationship between different immunosuppressive regimens and the antibody-mediated neutralizing capacity against different SARS-CoV-2 variants is visualized in Appendix A.

### 3.5. Machine Learning Model Identifying Correlates of SARS-CoV-2 Breakthrough Infection

To identify additional possible contributing factors to the risk of SARS-CoV-2 breakthrough infection, we applied a machine learning model based on antibody levels, neutralizing activity, immunosuppressant regimens, time since transplantation, and age.

In total, 5 out of 16 features were selected using elastic net regression, followed by a Bootstrap Forest model with 5-fold cross-validation. Based on these five features alone, the model yielded a high prediction rate for SARS-CoV-2 breakthrough infection with an AUC of 97% in the training set (Figure 5A) (the model correctly classified 86 negatives (true negatives) and 9 positives (true positives), while it misclassified 1 negative as positive (false positive) and 11 positives as negative (false negative). When applying the same model in an independent test set, the AUC was 78% (Figure 5B) (the model correctly classified 22 negatives and 1 positive, with 2 negatives mis-labeled as positive and 3 positives mis-labeled as negative.

The training sensitivity accordingly was (True Positive Rate, TPR) = 0.45; training specificity (True Negative Rate, TNR) = 0.99; test sensitivity = 0.25; and test specificity = 0.92. The drop in sensitivity from training to validation, along with decreases in the AUC, indicates moderate overfitting in detecting true positives, whereas the specificity remains high in both sets.

Only one patient had missing data. Performing list-deletion did not alter the result compared with that of the imputed model.

Age and neutralizing activity against SARS-CoV-2 Spike (WT), Delta, and B2 Omicron variants together accounted for more than 99% of the variation (Figure 5C).

## 4. Discussion

In this observational study, we analyzed the antibody response to a third SARS-CoV-2 mRNA vaccine dose and identified the correlates of protection against SARS-CoV-2 breakthrough infection in KTRs. A SARS-CoV-2-specific IgG antibody response was detected in 74% of the KTRs after a third vaccine dose with an approximately ten-fold increase in median SARS-CoV-2 IgG levels among those who responded. These results are consistent with previous studies measuring IgG seroconversion rates in KTRs and other SOTRs [16,17,18,19], showing seropositivity for anti-RBD and anti-S IgG after a third vaccine dose in 77% and 35%, respectively [20]. One systematic review and meta-analysis found a pooled seroconversion rate of 55% in SOTRs after a third vaccine dose [21]. Two other systematic reviews and meta-analyses have shown lower post-booster seropositivity rates of 41–50% in KTRs [22,23]. This may be explained by differences in immune assays, cut-off values, and sampling times.

The neutralizing antibody activity was significantly lower against all the Omicron variants when compared with that against the WT strain (SARS-CoV-2 Spike), while it remained relatively preserved against the Alpha, Beta, and Delta variants. Similar results have been observed in other studies showing reduced neutralizing activity against the Omicron variant B.1.1.529 compared with that against the WT strain [24,25], and the proportion of neutralizing antibodies has been previously shown to be lower against Omicron B.1.1.529 and BA.1 compared with that against WT and Delta variants [26,27,28]. In our study, neutralizing activity was lower against all the five Omicron variants analyzed (BA.1+R346K, BA.1, BA.1+LA25R, BA.2, and BA.3) compared with that against the WT strain. Our findings include additional Omicron variants, providing additional information regarding the neutralizing antibody response to a third vaccine dose in KTRs [24,25,26,27,28,29].

Interestingly, KTRs who experienced a SARS-CoV-2 breakthrough infection had significantly lower neutralizing activity against most viral strains after the third vaccine dose. This is in line with Al Jurdi et al. reporting that 6% of the KTRs developed a SARS-CoV-2 breakthrough infection after a third vaccine dose, none of whom had neutralizing antibodies against Omicron [26]. In another study, only a few KTRs with a symptomatic SARS-CoV-2 breakthrough infection had detectable neutralizing antibodies against Delta and Omicron BA.1 and BA.2 [29]. Dimeglio et al. reported protective antibody levels against Omicron BA.1 and BA.2 in healthcare workers, some of whom had been previously infected by SARS-CoV-2 [30]. In KTRs, a high anti-RBD IgG titer response prior to infection was predictive of protection 40 days after infection [31]. Although others found no significant difference in median anti-spike IgG titers between infected and uninfected KTRs [32], our results support the importance of the neutralizing abilities of the circulating antibodies in COVID-19 prevention.

KTRs on triple immunosuppressive therapy had significantly higher rates of SARS-CoV-2 breakthrough infections compared with those on mono- or dual-immunosuppressive regimens. Immunosuppressive drugs are known to reduce SARS-CoV-2 mRNA vaccine effectiveness, particularly systemic corticosteroids and rituximab in patients with immune-mediated inflammatory diseases [33]. Triple therapy, high-dose corticosteroids, and antimetabolite treatment are recognized risk factors for diminished serological responses in fully vaccinated SOTRs [6,8]. Additionally, antimetabolite use has been associated with poor serological response to a third vaccine dose in KTRs [25], and multivariate analyses have identified antimetabolite and steroid treatments as independent predictors of poor IgG and T-cell responses, respectively [32]. In our analysis, we stratified by immunosuppressant regimen rather than assessing individual drugs, as the treatment choices depend on multiple factors, including time since transplantation, comorbidities, underlying kidney disease, and medication side effects. Furthermore, immunosuppressant drugs are generally administered in combination in KTRs making it challenging to isolate the effects on vaccine response. To simplify this challenge, and while keeping the number of variables to a minimum, we chose to compare the triple immunosuppressive regimen with dual- and monotherapy as well as with rituximab administration less than one year prior to the first vaccine dose. A limitation to this method, however, is that it introduces a heterogenicity in the immunosuppressive regimens evaluated, which could have potential differential impacts on immune responses.

While low SARS-CoV-2 IgG antibody levels were not significantly associated with SARS-CoV-2 breakthrough infection risk, they remain linked to antibody-mediated neutralizing activity, immunosuppressant regimen, and age, among other factors, making it difficult to draw a conclusion from single-variable analyses in this patient population. We used our multivariate machine learning model to better incorporate these interrelated factors and identify the most robust correlating factors for SARS-CoV-2 breakthrough infection. Among the 16 features included in the model, age and neutralizing activity against WT (SARS-CoV-2 Spike), Delta, and the BA.2 Omicron variant emerged as the most important factors (Figure 5), while IgG levels were not significantly correlated to SARS-CoV-2 breakthrough infection.

This study has several other limitations. First, we cannot avoid a possible under-detection of asymptomatic or subclinical infections. To minimize the impact of hybrid immunity, we excluded patients with a prior positive SARS-CoV-2 PCR test and those who had received monoclonal antibody treatment before plasma sampling. However, we cannot entirely rule out subclinical infections that were not detected by PCR testing. Nonetheless, all the KTRs were systematically tested during the study period and were advised to contact their nephrology department or primary physician in the event of COVID-19 symptoms or a positive rapid SARS-CoV-2 antigen test, making undetected or unreported SARS-CoV-2 infections less likely. Second, we did not measure the long-term durability of IgG or neutralizing responses, although they play a role in the immunogenicity of the vaccines and risk of SARS-CoV-2 breakthrough infection [18,25]. It would be beneficial to have larger studies examining this aspect. Third, the sample size and geographic restriction to three Danish centers may affect generalizability. Also, there was heterogenicity in the vaccine type; although most KTRs received BNT162b2 and only four received the mRNA-1273 vaccine, they were analyzed together. It would be beneficial to analyze subgroups of vaccine types prospectively.

Last, we did not investigate cellular immune responses to the third vaccine dose, although they are also associated with protection against infection. In the literature, increased levels of SARS-CoV-2 spike-specific interferon gamma (IFN-γ)-producing T-cells have been observed in KTRs after a third SARS-CoV-2 mRNA vaccine dose [27,32] though they remain significantly lower than in healthy controls [16]. In contrast, Kemlin et al. did not measure a significant increase in T-cell responses after a third vaccine dose; however, they found a correlation with the neutralizing antibody response, suggesting overlapping information for these parameters. They identified SARS-CoV-2 S2-specific IFN-γ responses, RBD-binding IgG avidity, and neutralizing antibodies as correlates of protection against symptomatic SARS-CoV-2 breakthrough infection in KTRs. They further reported that, although age and antimetabolite therapy were associated with an altered vaccine response, they were not associated with SARS-CoV-2 breakthrough infections. In their multivariate analyses, the strongest predictor of protection was neutralizing antibody levels, followed by S2-specific IFN-γ response [29]. In line with these results, we identified neutralizing antibodies as strong correlates of protection; however, we did in fact also identify age as a correlating factor in our multivariate machine learning model. We did collect peripheral blood cells, but it was not possible to measure cellular immune response due to logistics. The challenge is that it is difficult to standardize and reproduce T-cell results in different laboratories, unless commercial kits or WHO-standardized assays exist that can be used broadly. However, we believe that T-cell mediated immunity assessment is crucial for the comprehensive evaluation of vaccine efficacy in KTRs, and we suggest including this in future prospective studies.

A strength of this study is that the timing of plasma sampling occurred just before a major surge in Omicron-driven COVID-19 cases in Denmark (Appendix A). SARS-CoV-2 breakthrough infections were recorded until the end of February 2022, coinciding with the peak of this wave. Therefore, we believe our findings predominantly reflect the vaccine response rather than prior exposure to these variants. With emerging SARS-CoV-2 variants, it is challenging to exclusively use IgG titers as protective measures, although they do reflect immune competence. Neutralizing antibody activity is a functional measure of the pool of different types of antibodies, and it can be a relevant supplement in assessing vaccine-derived SARS-CoV-2 immunity in KTRs, especially in outbreaks with SARS-CoV-2 variants. This study brings a deeper understanding of vaccine immunogenicity to different SARS-CoV-2 subvariants and correlating factors to SARS-CoV-2 breakthrough infection.

In conclusion, this study highlights that, while a third SARS-CoV-2 mRNA vaccine dose significantly enhances humoral responses in KTRs, it may confer insufficient protection against emerging variants such as Omicron, particularly in those under intensive immunosuppression. These findings underscore the need for personalized vaccination strategies.

## Figures and Tables

**Figure 1 vaccines-13-00777-f001:**
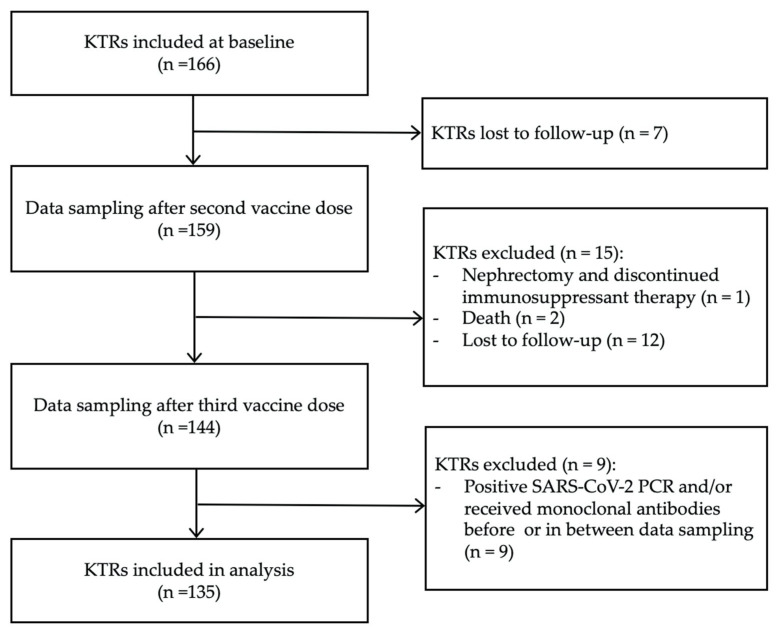
Flowchart illustrating inclusion and exclusion of study participants.

**Figure 2 vaccines-13-00777-f002:**
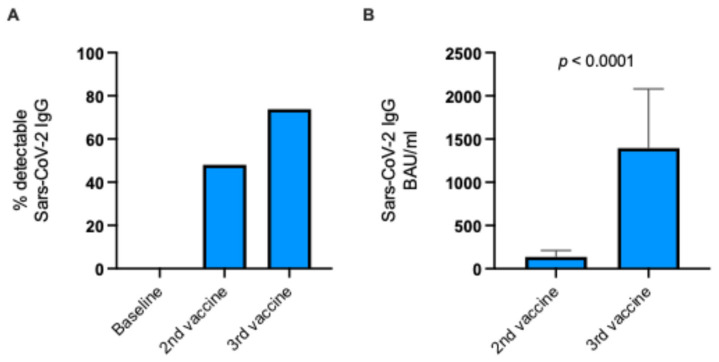
Immune response to a third SARS-CoV-2 mRNA vaccine dose in KTRs. (**A**) Percentage of KTRs with detectable SARS-CoV-2 IgG following second and third vaccine doses. (**B**) Median SARS-CoV-2 IgG levels following second and third vaccine doses for KTRs with detectable antibodies (Mann–Whitney test). KTRs, kidney transplant recipients; BAU/mL, binding antibody units per mL.

**Figure 3 vaccines-13-00777-f003:**
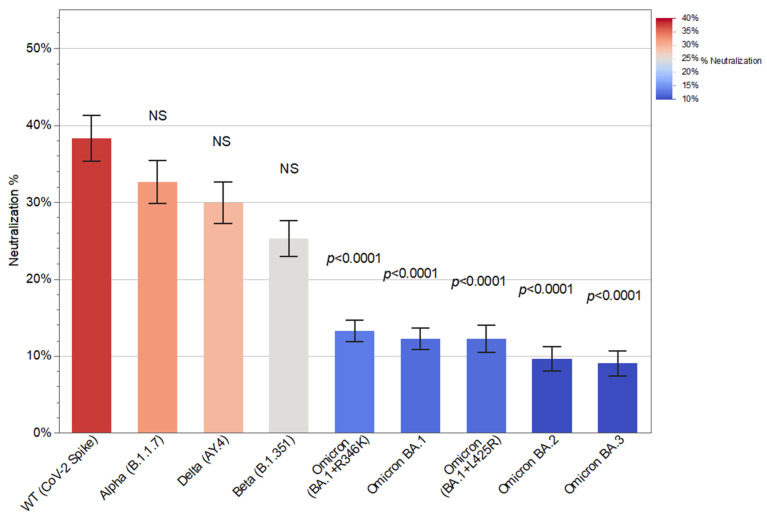
Antibody-mediated neutralization (%) against different variants of SARS-CoV-2. Serum samples were obtained from kidney transplant recipients (KTRs) after the third SARS-CoV-2 mRNA vaccine dose, and antibody neutralization activity was measured using an angiotensin-converting enzyme 2 (ACE2) pseudo-neutralization assay. For each of the variants (Alpha, Delta, Beta, and Omicron), the *p*-values reflect comparison with the wild-type (WT) strain (Kruskal–Wallis followed by Dunn’s) (mean; SEM).

**Figure 4 vaccines-13-00777-f004:**
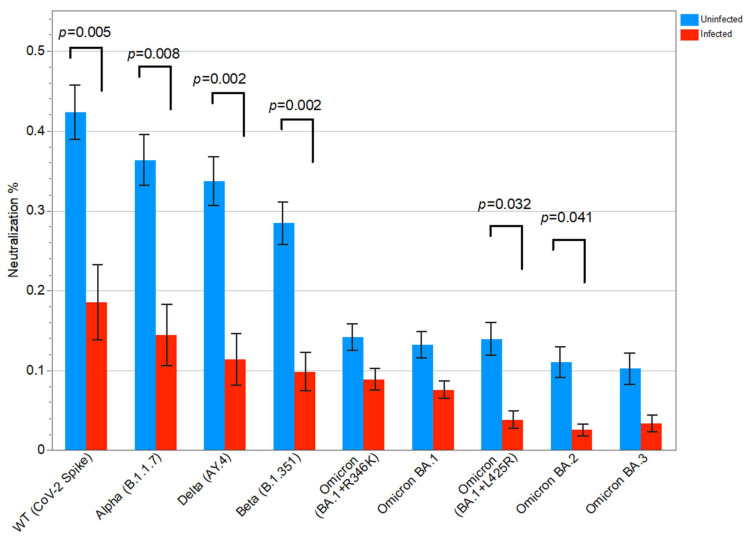
Antibody-mediated neutralization (%) compared between kidney transplant recipients (KTRs) with and without SARS-CoV-2 breakthrough infection using the Mann–Whitney test for each variant of SARS-CoV-2. The *p*-values not shown were non-significant (mean; SEM); WT, wild-type.

**Figure 5 vaccines-13-00777-f005:**
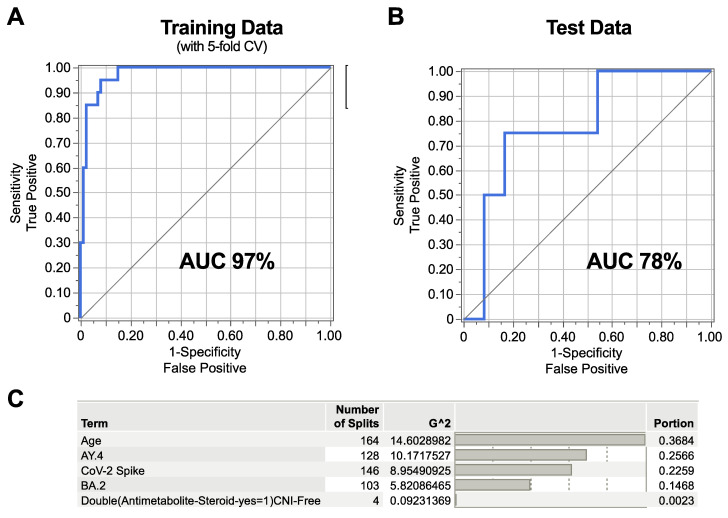
Machine learning model identifying the correlates of SARS-CoV-2 breakthrough infection. An elastic net model was used followed by Bootstrap Forest model, both using 5-fold cross-validation. The ROC curves depict the ability of the model to identify the correlates of SARS-CoV-2 breakthrough infection for (**A**) training data and (**B**) test data, based on the five selected features. (**C**) The table shows the contribution of each of the features to the model. CV, cross-validation; AUC, area under the curve; CNI, calcineurin inhibitor; AY.4, Delta SARS-CoV-2 variant; BA.2, Omicron SARS-CoV-2 variant.

**Table 1 vaccines-13-00777-t001:** Clinical characteristics of KTRs according to humoral response 6 weeks after a third SARS-CoV-2 mRNA vaccine dose.

Characteristic	All KTRs (n = 135)	Seropositive After Third Vaccine Dose (n = 100)	Seronegative After Third Vaccine Dose (n = 35)
Female, n (%)	55 (41%)	38 (38%)	17 (49,6%)
Age, median years (IQR)	55 (47–67)	54 (46–67)	61 (51–68)
Vaccine type (%)			
3 × BNT162b2 mRNA	131 (97%)	98 (98%)	33 (94.3%)
3 × mRNA1273	4 (3%)	2 (2%)	2 (5.7%)
Time between transplantation and first vaccine dose, years (IQR)	6.29 (2.85–11.98)	7.64 (3.97–13.18)	3.59 (0.71–6.54)
First transplant, n (%)	112 (83%)	81 (81%)	31 (88.6%)
Type of immunosuppressive regimen, n (%)			
Triple therapy (CNI, antimetabolite, steroids)	64 (47%)	38 (38%)	26 (74%)
Steroid-free dual therapy *	40 (29.6%)	33 (33%)	7 (20%)
Antimetabolite-free dual therapy **	14 (10.4%)	13 (13%)	1 (2.9%)
CNI-free dual therapy ***	5 (3.7%)	4 (4%)	1 (2.9%)
Rituximab < 1 year prior to first vaccine dose	6 (4.4%)	2 (2%)	4 (11.4%)

* Steroid-free dual-therapy regimens predominantly include the combinations of calcineurin inhibitor (CNI) + antimetabolite or sirolimus + antimetabolite. ** Antimetabolite-free dual-therapy regimens predominantly include CNI + prednisolone or sirolimus + prednisolone. *** CNI-free regimens predominantly include sirolimus + antimetabolite or sirolimus + prednisolone. A Venn diagram showing all combinations can be found in Appendix A. KTRs, kidney transplant recipients; n, Number; IQR, Interquartile Range.

**Table 2 vaccines-13-00777-t002:** Immunosuppressive regimen and risk of SARS-CoV-2 breakthrough infection.

Immunosuppressive Regimen	All KTRs n (%)	Risk of SARS-CoV-2 Breakthrough Infection, OR	Lower 95%	Upper 95%	Fisher’s Exact Test
Triple therapy (CNI + antimetabolite + steroid)	64 (47.4%)	2.63	1.04	6.64	**0.044**
Steroid-free dual therapy	40 (29.6%)	0.57	0.20	1.65	0.34
Antimetabolite-free dual therapy	14 (10.4%)	0.75	0.16	3.59	1.00
CNI-free dual therapy	5 (3.7%)	0.00	N/A	N/A	0.59
Rituximab < 1 year prior to vaccination	6 (4.4%)	0.92	0.10	8.27	1.00

CNI, calcineurin inhibitor; n, number; N/A, not applicable; OR, odds ratio.

## Data Availability

Data used for analyses will be available upon reasonable request.

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
