# Peer review of "Correlates of SARS-CoV-2 Breakthrough Infections in Kidney Transplant Recipients Following a Third SARS-CoV-2 mRNA Vaccine Dose"

_vaccines, 2025, doi:10.3390/vaccines13080777_

Round 1
Reviewer 1 Report
Comments and Suggestions for Authors
This is an interesting manuscript, reporting the antibody response of SARS-CoV2 among Kidney transplantation patients. Although the pandemic has ended, the result and what we learned from this study are beneficial to researchers and readers. To conduct research among up to 130+ kidney transplant patients was not easy.
I have a few comments/suggestions as follows:
- The timeline of this study was not so clear in the methodology and result. Please add some details about the year of research, since COVID19 pandemic was highly dynamic in many aspects such as virus variants, infectivity and severity of virus.
- Types of immunosuppressive medication clearly impact the antibody response. Do author have some details about these two categories in Table1 i.e. “Steroid-free Dual therapy” and “Antimetabolite-free Dual therapy”.
- Since this was a multicenter study conducted in 3 medical centers in Denmark. Is there any significant difference between immunosuppressive medication used after kidney transplant in each center. Do they have their own practice guideline, or it is relatively the same guideline among these three centers.
- Please mention the strengths and limitations of this research in the discussion as well.
Author Response
Comment 1: The timeline of this study was not so clear in the methodology and result. Please add some details about the year of research, since COVID19 pandemic was highly dynamic in many aspects such as virus variants, infectivity and severity of virus. Reply 1: We agree on this point. We have, accordingly, modified the section 2.1 “Participants” (page 2) in the methods to emphasize this point, including study period, recruitment of participants, and times of vaccination and sampling. We also corrected the date of inclusion from January 11, 2021, to January 15, 2021. We did not register virus variants systematically, as they were tested on different locations, but we have addressed this in the last part of method section 2.1 “Participants” (page 2) as well as 2.6 “National COVID-19 Infection Rates”(page 5). In both Figure S1 and S2, we modified the figure legend, highlighting that they illustrate approximate times of vaccine dose administrations and data sampling of our study cohort. Comment 2: Types of immunosuppressive medication clearly impact the antibody response. Do author have some details about these two categories in Table1 i.e. “Steroid-free Dual therapy” and “Antimetabolite-free Dual therapy”. Reply 2: Thank you for pointing this out. We agree with this comment. Therefore, we have added more text describing details on combinations of immunosuppressive drugs. We have added the following text to the table legend of Table 1, and in the main text in the result section 3.1 “Clinical and Demographic Characteristics“ (page 5): “The steroid-free dual therapy regimens predominantly include the combinations of calcineurin inhibitor (CNI) + antimetabolite or sirolimus + antimetabolite. Antimetabolite-free dual therapy regimens predominantly include CNI + prednisolon or sirolimus + prednisolon. CNI-free regimens predominantly include sirolimus + antimetabolites or sirolimus + prednisolon. A Venn diagram showing all combinations can be found in Figure S3.” Comment 3: Since this was a multicenter study conducted in 3 medical centers in Denmark. Is there any significant difference between immunosuppressive medication used after kidney transplant in each center. Do they have their own practice guideline, or it is relatively the same guideline among these three centers. Reply 3: The reviewer raises an important question. There are no general differences in guidelines between the three centers. What affects differences in immunosuppressive drugs the most is time since transplantation, comorbidities, underlying kidney disease, and medication side effects among others, more than a center-depended factor. Comment 4: Please mention the strengths and limitations of this research in the discussion as well Reply 4: Agree. We have, accordingly, revised the discussion and highlighted strengths, such as the timing of plasma sampling before a big Omicron surge in Denmark and that the recording of SARS-CoV-2 breakthrough infections was coinciding with the peak of the wave. Therefore, we believe, that our findings predominantly reflect the vaccine response rather than prior exposure to these variants. This can be found in discussion (page 12, paragraph 8). We have also included limitations, including lack of T-cell mediated immunity assessment, absence of long-term follow-up on antibody and neutralizing responses durability, and heterogenicity in immunosuppressive regimens, amongst others. This can be found in the discussion (page 12, paragraphs 4, 6, and 7).Reviewer 2 Report
Comments and Suggestions for Authors
This study evaluated a clinically significant concern: the suboptimal immune response to SARS-CoV-2 vaccination among kidney transplant recipients (KTRs), a population characterized by chronic immunosuppression. The findings demonstrate that administration of a third dose of SARS-CoV-2 mRNA vaccine significantly enhanced anti-spike IgG antibody titers, exhibiting approximately a ten-fold increase. Nevertheless, neutralizing activity against Omicron variants remained markedly limited. Breakthrough infections were predominantly observed in individuals with reduced neutralizing antibody titers, particularly those receiving triple immunosuppressive therapy. Among all variables assessed, age and the ability to neutralize key SARS-CoV-2 variants were the strongest predictors of breakthrough infection risk.
General comments
The manuscript presents a well-structured and timely investigation addressing a critical issue in transplant immunology and vaccine response. The author had adequately summarized prior literature, and the MS is logically organized. However, to enhance scientific precision and reproducibility, several revisions are recommended:
Specific recommendations::
- Scientific language and terminology:
- Ensure consistent and specific terminology throughout the MS (e.g., use “third SARS-CoV-2 mRNA vaccine dose” instead of “third vaccine booster”, and replace vague expressions such as “overall ten-fold increase” with quantitative descriptors like “approximately ten-fold increase”
- Define abbreviations at firs use and maintain consistency (e.g., KTRs, SARS-CoV-2, IgG, Nab).
- Other specific observations that needed to be addressed:
- KEYWORDS:COVID-19, Kidney Transplant, vaccines, machine learning predictors, transplantant immunology, immunosuppression. (The title words should not be repeated in Keywords).
- ABSTRACT
Background: Kidney transplant recipients (KTRs) exhibit a significantly diminished immune response to Severe Acute Respiratory Syndrome Coronavirus-2 (SARS-CoV-2) vaccines compared with to the general population, primarily due to ongoing immunosuppressive therapy. This study investigated evaluated the immunogenicity immune response to of a third dose ofmRNA-based SARS-CoV-2 vaccine dose in KTRs and assessed antibodies as correlates the association between antibody responses and of protection against along with other potential predictors of SARS-CoV-2 breakthrough infection. Additionally, clinical and immunological correlates of post-vaccination SARS-CoV-2 infection were examined.
Methods: A total prospective cohort of 135 KTRs received a third vaccine dose approximately six months after following the second dose. Plasma samples were collected at baseline (before the first dose pre-vacination), six months after the second dose, and six weeks after following the third dose. Humoral responses were assesed using SARS-CoV-2-specific IgG titters and virus neutralization assays against wild-type (WT) and variant strains, including multiple Omicron sublineages.
Results: Following After the third vaccine dose, 74% of /KTRs had detectable SARS-CoV-2 specific IgG antibodies, compared with only to 48% after following, the second dose. Mean IgG titters increased approximately SARS-CoV-2 IgG antibody levelstenfold post -booster. after the third dose. However, Despite this increase, neutralizing activity against several Omicron variants was remained significantly lower than that against the wild-type WT strain. KTRs who subsequently experienced SARS-CoV-2 breakthrough infection had lower demonstrated reduced neutralizing antibody activity against across all variants tested. Additionally, Those individuals receiving on triple immunosuppressive therapy had a significantly higher risk of SARS-CoV-2 breakthrough infection compared with to those on dual or single monotherapy. In a A multivariate machine learning analysis modelindentified age, along with and neutralizing activity against WT, Delta, and the BA.2 Omicron BA.2 variant, were the strongest as the most robust predictors of SARS-CoV-2 breakthrough infection.
Conclusion: A third dose of an mRNA vaccine dose -based significantly improves SARS-CoV-2 – specific IgG levels in KTRs; however, the neutralizing response activity and against Omicron variants remains diminished suboptimal. KTRs with lowerDiminished neutralizing capacity or those receiving triple and intensified immunosuppression are key determinants at significantly higher risk of breakthrough SARS-CoV-2 infection in this immunocompromised population.
- Methods section:
- Participants:
- Clearly describe the inclusion/exclusion criteria
- Detail the recruitment chronologically, including the timeline of vaccination and biological sampling.
- Specify the timing of samples collection in relation to vaccination and clinical event
- SARS-CoV-2 Antibody Measurement:
To ensue methodological transparency:
- Provide the timing the timing between blood collection and sample processing, as well as storage conditions (e.g., temperature, duration).
- Specify centrifugation parameters (e.g., g-force, duration).
- Report assay performance characteristics:
- Analytical sensitivity and specificity.
- Intra- and inter-assay variability.
- Calibration procedure using WHO standards.
- Internal quality control procedures.
- Whether laboratory analyses were blinded to clinical outcomes.
- Machine Learning Models:
- Clarify the development stages: data processing, features selection, normalization, and scaling.
- Specify the software used.
- Discuss model performance metrics and interpretability (future importance rankings).
- National COVID-19 Infection Rates and Data Integration:
- Provide specific dates for data acquistion and granularity of infection/variant datasets (daily vs weekly).
- Cite the data sources and versions (e.g., national dashboards).
- Describe how the national epidemiological data was temporally aligned with the clinical study timeline.
- Study Design Clarification and Limitations:
- Emphasize the novelty and clinical relevance of focusing in neutralizing antibody response post-third mRNA dose in a highly immunocompromised cohort.
- Clearly articulate the limitations:
- Lack of T-cell mediated immunity assessment, which is crucial for comprehensive evaluation of vaccine efficacy in KTRs.
- Possible under-detection of asymptomatic or subclinical infections.
- Limited sample size and geographic restriction in three Danish centres, which may affect generalizability.
- Heterogenicity in immunosuppressive regimens, with potential differential impacts on immune responses.
- Absence of long-term follow-up data on antibody (humoral and neutralizing responses) durability.
- Conclusion
This study highlights that while a third SARS-CoV-2 mRNA vaccine dose significantly enhances humoral responses in KRTs, it may confer insufficient protection against emerging variants such as Omicron, particularly in older individuals and those under intensive immunosuppression. These findings underscore the need for personalized vaccination strategies.
Comments on the Quality of English Language
There are English language errors that will need to be addressed
Author Response
Comment 1: Specific recommendations:
- Scientific language and terminology:
- Ensure consistent and specific terminology throughout the MS (e.g., use “third SARS-CoV-2 mRNA vaccine dose” instead of “third vaccine booster”, and replace vague expressions such as “overall ten-fold increase” with quantitative descriptors like “approximately ten-fold increase”
- Define abbreviations at firs use and maintain consistency (e.g., KTRs, SARS-CoV-2, IgG, Nab)
Reply 1: We agree with the reviewer on this. We have, accordingly, changed “third vaccine booster” to “third SARS-CoV-2 mRNA vaccine dose” throughout the manuscript and ensured consistency. We have rephrased “overall ten-fold increase” with “approximately ten-fold increase” and made sure that abbreviations were defined as suggested. This last correction can be found in the results of the abstract, result section 3.2 “Vaccine-Induced SARS-CoV-2 Antibody Response” (page 7), and discussion (page 10, paragraph 1).
Comment 2: KEYWORDS:
COVID-19, Kidney Transplant, vaccines, machine learning predictors, transplantant immunology, immunosuppression. (The title words should not be repeated in Keywords). Reply 2: Agree. We have, accordingly, changed the keywords to your suggestions. Furthermore, we removed “vaccines” as a keyword, as it is also a part of the title (page 2). Comment 3: Abstract revision suggestions Reply 3: Thank you for your suggested revisions to the abstract. We have included them in the revised manuscript. Comment 4: Methods section:- Participants:
- Clearly describe the inclusion/exclusion criteria
- Detail the recruitment chronologically, including the timeline of vaccination and biological sampling.
- Specify the timing of samples collection in relation to vaccination and clinical event
Reply 4: Thank you for pointing this out. We agree with this comment.
Regarding inclusion/exclusion criteria, we have described in more details and added the following to the end of the method section 2.1 “Participants” (page 3): “We also excluded KTRs who were lost to follow-up including death, or discontinued immunosuppression therapy due to nephrectomy. All KTRs included followed the vaccination schedule and had plasma samples collected as described.”
We agree on the two other points on recruitment and timeline. We have, accordingly, modified section 2.1 “Participants” (page 2 and 3) in the methods to emphasize this point, including highlighting the study period, added details on recruitment of participants, and times of vaccination and sampling. We also corrected the date of inclusion from January 11, 2021, to January 15, 2021.
In both Figure S1 and S2, we modified the figure legend, highlighting that they illustrate ”approximate times of vaccine dose administrations and data sampling of our study cohort” as icons are placed on the graphs, and approximately illustrates the study timeline. The specific information can be found in section 2.1 “Participants” (page 2 and 3), as mentioned.
Comment 5: SARS-CoV-2 Antibody Measurement:
To ensue methodological transparency:
- Provide the timing the timing between blood collection and sample processing, as well as storage conditions (e.g., temperature, duration).
- Specify centrifugation parameters (e.g., g-force, duration).
- Report assay performance characteristics:
- Analytical sensitivity and specificity.
- Intra- and inter-assay variability.
- Calibration procedure using WHO standards.
- Internal quality control procedures.
- Whether laboratory analyses were blinded to clinical outcomes.
Reply 5: Thank you for pointing this out. We agree, therefore, we have added more information regarding your suggested points to the method section 2.2 “SARS-CoV-2 Antibody Measurement” (page 4) accordingly, by adding the following text: “The assay's performance characteristics, as reported by Bonelli et al. [10] include a clinical sensitivity of 98.7% (≥15 days after a positive PCR result) and a specificity of 99.5% (95% confidence interval (CI) 99.0%-99.7%). Intra-assay coefficient of variation (CV) is 1,6-5,1% and inter-assay CV is up to 6% according to the kit insert.Calibrator concentrations, expressed in BAU/mL, have been standardized by the manufacturer against the First WHO International Standard for anti-SARS-CoV-2 immunoglobulin (NIBSC code: 20/136). All samples were analyzed by experienced laboratory technicians in accordance with the manufacturer´s protocol, including the daily testing of the LIAISON® SARS-CoV-2 TrimericS IgG internal control samples. The laboratory analysis was blinded to clinical outcomes.”
Comment 6: Machine Learning Models:
- Clarify the development stages: data processing, features selection, normalization, and scaling.
- Specify the software used.
- Discuss model performance metrics and interpretability (future importance rankings).
Reply 6: We agree with the reviewer that this information can be more detailed. Accordingly, and in alignment with other reviewer suggestions, we have modified the manuscript with the following:
in the methods section “2.5 Machine Learning Models” (page 5):
“Data Processing: Raw data were analyzed using JMPPro 16 (SAS Institute). Features with > 20 % missingness were excluded; remaining missing values were imputed with k-nearest neighbors (k = 5).
Feature Selection: Elastic net, which applies a dual penalty on both the absolute values of coefficients and the sum of squared coefficients, enabling the algorithm to shrink coefficients toward zero was first used to prioritize important features. We loaded 16 features based on antibody levels, neutralizing activity for each variant, immunosuppressant regimens, time since transplantation and age.
On the training set (n = 107), we fit an elastic-net model (α = 0.5) via 5-fold cross-validation to identify predictive variables. The penalty λ minimizing mean CV error retained 5 nonzero terms for subsequent modeling in the Bootstrap Forest model. Early stopping was used.
Normalization and Scaling: All continuous predictors were standardized; categorical predictors were dummy-coded. Standardization ensured comparability both under the elastic-net penalty and for random-forest split criteria.”
in the results section “3.5 Machine Learning Models” (page 8 and 10):
“Five out of 16 features were selected using elastic net regression, followed by a Bootstrap Forest model with 5-fold cross-validation. Based on these five features alone, the model yielded a high prediction rate for SARS-CoV-2 breakthrough infection with an AUC of 97% in the training set (Figure 5A) (the model correctly classified 86 negatives (true negatives) and 9 positives (true positives), while it misclassified 1 negative as positive (false positive) and 11 positives as negative (false negative). . When applying the same model in an independent test set AUC was 78% (Figure 5B) (the model correctly classified 22 negatives and 1 positive, with 2 negatives mis‐labelled as positive and 3 positives mis‐labelled as negative.
Training Sensitivity accordingly was (TPR) = 0.45, training Specificity (TNR) = 0.99, test Sensitivity = 0.25, and test Specificity = 0.92
The drop in sensitivity from training to validation, along with decreases in AUC, indicates moderate overfitting in detecting true positives, whereas specificity remains high in both sets.”
Comment 7: National COVID-19 Infection Rates and Data Integration:
- Provide specific dates for data acquistion and granularity of infection/variant datasets (daily vs weekly).
- Cite the data sources and versions (e.g., national dashboards).
- Describe how the national epidemiological data was temporally aligned with the clinical study timeline.
Reply 7: Thank you for this comment. We agree, therefore, we have added more details to the method section 2.6 ”National COVID-19 Infection Rates” (page 5) including the specific date of obtaining data of National COVID-19 infection rates and variants of concern and cited the data sources.
We also described how the national epidemiological data was temporally aligned with the clinical study timeline, by including the following text to method section 2.1 “Participants” (page 3): “Specific SARS-CoV-2 variants were not registered systematically, because the KTRs were tested at different locations. However, the plasma sampling before and after the third vaccine dose occurred when the Delta variant was dominating in Denmark which was also just before a major surge in Omicron-driven COVID-19 cases in Denmark, peaking in February/March 2022. We therefore assume, that SARS-CoV-2 breakthrough infections were predominantly Omicron variants. See methods 2.7 for more details.”
Comment 8: Study Design Clarification and Limitations:
- Emphasize the novelty and clinical relevance of focusing in neutralizing antibody response post-third mRNA dose in a highly immunocompromised cohort.
- Clearly articulate the limitations:
- Lack of T-cell mediated immunity assessment, which is crucial for comprehensive evaluation of vaccine efficacy in KTRs.
- Possible under-detection of asymptomatic or subclinical infections.
- Limited sample size and geographic restriction in three Danish centres, which may affect generalizability.
- Heterogenicity in immunosuppressive regimens, with potential differential impacts on immune responses.
- Absence of long-term follow-up data on antibody (humoral and neutralizing responses) durability.
Reply 8: Thank you for pointing this out.
To emphasize strengths, novelty, and clinical relevance of the study, we have added the following paragraph to the discussion (page 12, paragraph 8): “A strength to this study is that the timing of plasma sampling occurred just before a major surge in Omicron-driven COVID-19 cases in Denmark (Figures S1 and S2). SARS-CoV-2 breakthrough infections were recorded until the end of February 2022, coinciding with the peak of this wave. Therefore, we believe our findings predominantly reflect the vaccine response rather than prior exposure to these variants. With emerging SARS-CoV-2 variants, it is challenging to exclusively use IgG titers as protective measures, although they do reflect immune competence. Neutralizing antibody activity is a functional measure of the pool of different types of antibodies, and it can be a relevant supplement in assessing vaccine-derived SARS-CoV-2 immunity in KTRs, especially in outbreaks with SARS-CoV-2 variants. This study brings a deeper understanding of vaccine immunogenicity to different SARS-CoV-2 subvariants and correlating factors to SARS-CoV-2 breakthrough infection.”
We agree on all the suggested limitations.
To increase the logical flow and clearly articulate limitations, we rearranged the text to assemble them closer together and highlighted them as limitations. We added the following suggestions:
We added your suggestion to mention the absence of long-term follow-up data on antibody durability and cited references to underline this (page 11, paragraph 6): “Second, we did not measure long-term durability of IgG or neutralizing responses, although they play a role in the immunogenicity of the vaccines and risk of SARS-CoV-2 breakthrough infection. It would be beneficial with more studies examining this aspect or including it in larger studies”
We also agree that sample size, and geographic restriction is a limitation. In an optimal scenario we would have included regional and national centers. To emphasize this point, we have mentioned this as a limitation in the discussion (page 11, paragraph 6): “Third, the sample size and geographic restriction to three Danish centers may affect generalizability”
We also agree that heterogenicity in immunosuppressive regimens is a challenge. The drugs are generally administered in combination, inevitably making it challenging to isolate the effects of each specific drug on the vaccine response in this setting. Heterogenicity in these variables is creating more smaller subgroups, which is challenging to evaluate. As an attempt to simplify this challenge, and at the same time to keep the number of variables to a minimum, we chose to include the most common combinations, namely triple-immunosuppressive therapy and comparing this to duel- and monotherapy. On the other hand, there is this limitation you pointed out, on potentially impacting the measurements of the immune response. To emphasize this point, we rephrased this section and highlighted, that it is a limitation (page 11, paragraph 4): “To simplify this challenge, and while keeping the number of variables to a minimum, we chose to compare triple immunosuppressive regimen with dual- and monotherapy as well as rituximab administration less than one year prior to the first vaccine dose. A limitation to this method, however, is that it introduces a heterogenicity in the immunosuppressive regimens evaluated, which could have potential differential impacts on immune responses.”
T-cell mediated immunity is indeed an important analysis. We did collect peripheral blood cells, but it was not possible to measure cellular immune response due to logistics. We added more context and references to the discussion and updated this paragraph in the discussion (page 11-12, paragraph 7), including adding the explanation on why we did not perform analyses on cellular immunity and added the following text: ”We did collect peripheral blood cells, but it was not possible to measure cellular immune response due to logistics. The challenge is, that it is difficult to standardize and reproduce T-cell results in different laboratories, unless commercial kits or WHO standardized assays exist, that can be used broadly. However, we believe that T-cell mediated immunity assessment is crucial for comprehensive evaluation of vaccine efficacy in KTRs, and we suggest including this in future prospective studies.”
Comment 9: Conclusion revision suggestions:
Reply 9: Thank you for the suggested revisions to the conclusion. We have included them in the revised manuscript. Discussion paragraph 9, page.
Reviewer 3 Report
Comments and Suggestions for Authors
This manuscript predicted and analyzed the factors associated with SARS-CoV-2 breakthrough infection in KTRs after received the third mRNA vaccine, demonstrating significant particle value for clinical research.
However, this manuscript contains the follwing issues:
1. Insufficient figure legends. The legends for Fig.3 requires more detailed annotations(Indicating the serum sample were obtained from the KRTs after the third mRNA vaccine.
2. Redundancy in result section. Section 3.3 and 3.4 present overly concise content, which should be consolidated into a unified subsection to improve logical flow.
Author Response
Comment 1: Insufficient figure legends. The legends for Fig.3 requires more detailed annotations(Indicating the serum sample were obtained from the KRTs after the third mRNA vaccine.
Reply 1: Thank you for pointing this out. We agree with this comment. Therefore, we have revised the figure legend for Figure 3 by describing the method in more details including the timing of serum samples after the third SARS-CoV-2 mRNA vaccine dose. The change can be found in Figure 3 (page 7).
Comment 2: Redundancy in result section. Section 3.3 and 3.4 present overly concise content, which should be consolidated into a unified subsection to improve logical flow.
Reply 2: Agree. We have, accordingly, revised the result section and consolidated section 3.3 and 3.4 into one subsection regarding all our results on neutralizing IgG response to emphasize this point of having a more logical flow. This change can be found on page 7, where the paragraph is now changed to 3.3 “Neutralizing Antibodies to Different SARS-CoV-2 Variants and Associations with SARS-CoV-2 Breakthrough Infection”. We changed the following section numbers accordingly.
Reviewer 4 Report
Comments and Suggestions for Authors
The article "Predictors of SARS-CoV-2 Breakthrough Infections in Kidney 2
Transplant Recipients Following a Third SARS-CoV-2 mRNA 3
Vaccine Dose" proposed by Thygesen et al. is interesting but hardly innovative in this area. The link between protection and antibody levels has already been widely described in the literature (see the work of Dimeglio et al.), and the impact of different vaccine doses on transplant recipients has been discussed several times by Nassim Kamar. It would be more than judicious to include these various references in your article.
A few additional remarks:
1. "Predictors" is a strong word with very special significance in statistics. Please remove/modify
2. Since the vaccine is administered at variable intervals (e.g., two doses between 20 and 44 days), it is necessary to adjust for this variable, particularly in the machine learning model.
3. Please provide details on the composition of your test and training datasets, as well as the associated confusion matrices.
4. Not all patients received the same vaccination protocol. It would be useful to standardize your starting point by removing the four discordant patients.
Form: There is no need to exceed two decimal places for p-values.
Author Response
Comment 1: Vaccine Dose" proposed by Thygesen et al. is interesting but hardly innovative in this area. The link between protection and antibody levels has already been widely described in the literature (see the work of Dimeglio et al.), and the impact of different vaccine doses on transplant recipients has been discussed several times by Nassim Kamar. It would be more than judicious to include these various references in your article.
Reply 1: Thank you for your comments and we appreciate pointing out the work of Chloé Dimeglio and Nassim Kamar. Therefore, we have included more references in our article.
First, we have included the work of Dimeglio et al. including the following reference:
- Dimeglio et al. Antibody Titers and Protection against Omicron (BA.1 and BA.2) SARS-CoV-2 Infection. Vaccines. 2022.
We updated the text in the manuscript and added the following to the discussion (page 10, paragraph 3): “Dimeglio et al. reported protective antibody levels against Omicron BA.1 and BA.2 in health care workers, some of whom had been previously infected by SARS-CoV-2 [30]”. Additionally, we realized that our reference of Hovd et al. indeed also found protective antibody titers, and added the following in the same paragraph: ”In KTRs, high anti-RBD IgG titer response prior to infection was predictive of protection 40 days after infection [31].” We rephrased the following to: ”Although others found no significant difference in median anti-spike IgG titers between infected and uninfected KTRs [32], our results support the importance of neutralizing abilities of the circulating antibodies in COVID-19 prevention.” (page 10, paragraph 3).
Regarding Nassim Kamar, we have included the following references:
- Li et al. Factors Associated With COVID-19 Vaccine Response in Transplant Recipients: A Systematic Review and Meta-analysis. Transplantation. 2022.
- Kamar et al. Three Doses of an mRNA Covid-19 Vaccine in Solid-Organ Transplant Recipients. Letter to the editor. N Eng J Med. 2021
- Kamar et al. Anti‐SARS‐CoV‐2 spike protein and neutralizing antibodies at one and 3 months after 3 doses of SARS‐CoV‐2 vaccine in a large cohort of solid‐organ‐transplant patients. Am J Transplant. 2022.
We updated the discussion (page 10 paragraph 1) and added the following: “One systematic review and meta-analysis found a pooled seroconversion rate of 55% in SOTRs after a third vaccine dose [21].”
We also included a few more references of the work of Kamar et al. to the first part of the discussion regarding our results on IgG seroconversion after the third vaccine dose (page 10, paragraph 1), as well as to the discussion (page 11, paragraph 6) in the discussion regarding antibody durability.
Additional comments:
Comment 1: A few additional remarks: "Predictors" is a strong word with very special significance in statistics. Please remove/modify
Reply 1: Agree. We have, accordingly, changed this word into “correlates” and rephrased the sentences regarding our results, and the correction can be found throughout the manuscript text, including our title, abstract, aim and conclusions, to have a more neutral word.
Comment 2: Since the vaccine is administered at variable intervals (e.g., two doses between 20 and 44 days), it is necessary to adjust for this variable, particularly in the machine learning model.
Reply 2: Thank you for your comment. This is an important point the KTRs did receive the vaccine doses with varying intervals. Regarding antibody levels there was no correlation between the time intervals and levels. Also, when running ANCOVA we found no association. For the Elastic net model, adding time intervals did not alter the selected (non-zeroed) parameters in the bootstrap forest and accordingly not the final model. In conclusion, we do not find that the time between intervals affect the overall results, but we thank the reviewer for pointing this potential problem out.
Comment 3:Please provide details on the composition of your test and training datasets, as well as the associated confusion matrices.
Reply 3: We agree with the reviewer that the information on the machine learning models can be more detailed both in both methods description and result. Accordingly, and in alignment with other reviewer suggestions, we have modified the manuscript with the following:
in the methods section “2.5 Machine Learning Models” (page 5):
“Data Processing: Raw data were analyzed using JMPPro 16 (SAS Institute). Features with > 20 % missingness were excluded; remaining missing values were imputed with k-nearest neighbors (k = 5).
Feature Selection: Elastic net, which applies a dual penalty on both the absolute values of coefficients and the sum of squared coefficients, enabling the algorithm to shrink coefficients toward zero was first used to prioritize important features. We loaded 16 features based on antibody levels, neutralizing activity for each variant, immunosuppressant regimens, time since transplantation and age.
On the training set (n = 107), we fit an elastic-net model (α = 0.5) via 5-fold cross-validation to identify predictive variables. The penalty λ minimizing mean CV error retained 5 nonzero terms for subsequent modeling in the Bootstrap Forest model. Early stopping was used.
Normalization and Scaling: All continuous predictors were standardized; categorical predictors were dummy-coded. Standardization ensured comparability both under the elastic-net penalty and for random-forest split criteria.”
in the results section “3.6 Machine Learning Models” (page 8 and 10):
“Five out of 16 features were selected using elastic net regression, followed by a Bootstrap Forest model with 5-fold cross-validation. Based on these five features alone, the model yielded a high prediction rate for SARS-CoV-2 breakthrough infection with an AUC of 97% in the training set (Figure 5A) (the model correctly classified 86 negatives (true negatives) and 9 positives (true positives), while it misclassified 1 negative as positive (false positive) and 11 positives as negative (false negative). . When applying the same model in an independent test set AUC was 78% (Figure 5B) (the model correctly classified 22 negatives and 1 positive, with 2 negatives mis‐labelled as positive and 3 positives mis‐labelled as negative.
Training Sensitivity accordingly was (TPR) = 0.45, training Specificity (TNR) = 0.99, test Sensitivity = 0.25, and test Specificity = 0.92
The drop in sensitivity from training to validation, along with decreases in AUC, indicates moderate overfitting in detecting true positives, whereas specificity remains high in both sets.”
Comment 4: Not all patients received the same vaccination protocol. It would be useful to standardize your starting point by removing the four discordant patients.
Reply 4: Thank you for pointing this out. We included participants before knowing which vaccine they received. We also didn’t have any influence on the choice and timing on the vaccine doses, as it was determined nationally. There were changes in vaccination programs, i.e. the AstraZenica vaccine was phased out in this period, and we had anticipated more groups of vaccines and more KTRs in different groups. In our protocol we didn’t specify to exclude participants due to different types of vaccine. It would therefore be ethical problematic to exclude these data. If there had been more participants in different vaccine groups, we would have examined these subgroups. In this case, we hardly believe, that the result would be significantly different. However, we do believe that it is a relevant point. Therefore, we have added the following limitation to our discussion (page 11, paragraph 4): “Also, there was a heterogenicity in vaccine type, although most KTRs received BNT162b2 and only received 4 the mRNA-1273 vaccine, they were analyzed together. It would be beneficial to analyze subgroups of vaccine types prospectively.”
Comment 5: Form: There is no need to exceed two decimal places for p-values.
Reply 5: Thank you for pointing this out. We agree with this comment. Therefore, we have modified Figure 4 (page 8) and changed p-values to two decimal places. However, we also changed p-values to p<0.001 where relevant.
Round 2
Reviewer 2 Report
Comments and Suggestions for Authors
I appreciate the authors revisions to the manuscript. I no additional comments at this time and leave the editorial decision to the Editor.
Author Response
Comment 1: I appreciate the authors revisions to the manuscript. I no additional comments at this time and leave the editorial decision to the Editor.
Reply 1: Thank you very much for your comment.
Kind regards,
Miriam Viktov Thygesen
M.D.
Reviewer 4 Report
Comments and Suggestions for Authors
First of all, thank you for the quality of the responses provided to the previous comments.
Just one additional comment.
By imputing your missing data using the k-nearest neighbours method, you run the risk of accentuating the biases already existing in your initial data. It is preferable not to use any imputation method and to work on complete data.
Author Response
Comment 1: First of all, thank you for the quality of the responses provided to the previous comments.
Just one additional comment.
By imputing your missing data using the k-nearest neighbours method, you run the risk of accentuating the biases already existing in your initial data. It is preferable not to use any imputation method and to work on complete data.
Response 1: Thank you for the kind remark. We acknowledge the reviewer's concern regarding imputation.
Imputations can both accentuate or reduce biases, dependent on the dataset, the imputation method and ML method.
While Random Forest models (and other tree-based models) indeed in some cases are good at handling missing data, it is not the case for e.g. Elastic Net. As such, imputations are widely used in ML models, especially Elastic net.
Another alternative is of course, as the reviewer indicates, to leave out subjects with any missing values (often referred to as “listwise deletion”). It can in some cases be a good solution and reduce bias, but in other cases it can potentially introduce other biases; especially if you are not sure that the missing data are completely randomly missing.
In this specific case, we only have one patient with missing data (<1%), and one can argue that it hardly matters if you imputate or delete in this case, but in some cases it will.
To assess if there was a difference, we ran the algorithms again, leaving out the patient with missing data: the final result was hardly changed (AUC increased a bit from 97.4 to 97.6 in the training set and from 77.8 to 78.2 in the test set). In conclusion, the imputation in this case does not seem to alter the results in either direction; however, we would prefer to keep this method with the argument that it is the best total solution for this specific data set and ML methods used.
Overall, the reviewer's valid concern regarding the accentuation of existing biases through imputation is acknowledged. While our internal checks demonstrated that listwise deletion did not significantly alter the final model results, we believe our imputation strategy in this specific setting offers superior methodological soundness. It enables us to utilize our data more comprehensively and especially to account for any underlying non-random patterns within the missing data.
To clarify for the reader the amount of missingness, we have added the following to the results (result section 3.5, page 10, line 317): “Only one patient had missing data. Performing list-deletion did not alter the result compared to the imputed model.”